# CD146 Defines a Mesenchymal Stromal Cell Subpopulation with Enhanced Suppressive Properties

**DOI:** 10.3390/cells11152263

**Published:** 2022-07-22

**Authors:** Jean-Pierre Bikorimana, Wael Saad, Jamilah Abusarah, Malak Lahrichi, Sebastien Talbot, Riam Shammaa, Moutih Rafei

**Affiliations:** 1Department of Microbiology, Infectious Diseases and Immunology, Université de Montréal, Montréal, QC H3T 1J4, Canada; jean.pierre.bikorimana@umontreal.ca; 2Department of Pharmacology and Physiology, Université de Montréal, Montréal, QC H3T 1J4, Canada; wael.saad@umontreal.ca (W.S.); jamila_abusara@hotmail.com (J.A.); malak.lahrichi@umontreal.ca (M.L.); sebastien.talbot@umontreal.ca (S.T.); 3Canadian Centers for Regenerative Therapy, Toronto, ON M5R 1A8, Canada; 4IntelliStem Technologies Inc., Toronto, ON M5R 3N5, Canada; 5Molecular Biology Program, Université de Montréal, Montréal, QC H3T 1J4, Canada

**Keywords:** mesenchymal stromal cells, immunosuppression, CD146, mixed lymphocyte reaction, macrophages, efferocytosis, GVHD

## Abstract

Mesenchymal stromal cells (MSCs) are largely known for their immune-suppressive capacity, hence, their common use in the control of unwanted inflammation. However, novel concepts related to their biology, combined with the urgent need to identify MSC subpopulations with enhanced suppressive properties, drive the search for isolation protocols optimized for clinical applications. We show, in this study, that MSCs expressing high CD146 levels exhibit altered surface expression profiles of CD44 and secrete elevated levels of interleukin (IL)-6, amongst other factors. In addition, CD146^hi^ MSCs surpass the polyclonal parental populations in inhibiting alloreactive T cells in vitro, in both a soluble- and cell-contact-dependent manner. Despite the lack of CD146^hi^ MSC-mediated activation of peritoneal macrophages to release the suppressive factor IL-10 in vitro, their administration in animals with graft-versus-host disease alleviates inflammation and leads to 40% survival rate up to 7 weeks post-transplantation. This pronounced inhibitory property is driven by CD146-mediated in situ efferocytosis by myeloid cells. Altogether, this study provides the impetus to adopt an isolation protocol for MSCs based on a CD146 expression profile before their therapeutic use and suggests a major role played by CD146 as a novel “eat-me” signal, capable of enhancing MSC uptake by competent phagocytes.

## 1. Introduction

Due to their impressive pleiotropic potential, mesenchymal stromal cells (MSCs) are often perceived as the next-generation “Holy Grail” therapy for various illnesses. This is evidenced by their beneficial effect in the context of graft-versus-host disease (GVHD) [1], neurological disorders [2], and cardiovascular disease [3]. According to various studies, the therapeutic potency of MSCs is largely associated with their responsiveness to a variety of environmental cues, which, in return, regulate their paracrine effect on neighbouring cells [4,5]. Some of their described effects include modulating the apoptotic or angiogenic response of target cells, local tissue regeneration, and/or additional cross-communication with resident stem cells in the bone marrow (BM) [6,7,8,9,10,11,12]. Amongst the various mechanisms describing MSCs’ mode of action, Giri et al. recently reported the formation of a CCL2–CXCL12 chemokine complex in the secretome of BM-derived MSCs capable of binding and re-programming CCR2^+^ macrophages to secrete the suppressive factor IL-10 [13]. As a result, these IL-10-producing macrophages further amplify the anti-inflammatory cascade, resulting in colitis recovery [13]. Despite similar reports highlighting the importance of the secretome in mediating an anti-inflammatory paracrine effect, this type of suppressive mechanism remains largely questionable, as MSCs usually become undetectable shortly after their in vivo administration [14]. In fact, two recent key studies resolved this dogma by describing a model, whereby MSCs undergo efferocytosis mediated by resident phagocytes shortly after their in vivo administration [15,16]. Although both reports agreed on this concept, divergence remains as to whether host cytotoxic or alloreactive T cells are required to trigger MSC apoptosis prior to efferocytosis. This begs the question: is the secretome of MSCs involved in phagocyte recruitment or do MSCs express a given “turn-on” signal(s), leading to their uptake by endogenous phagocytes?

So far, preclinical studies have provided compelling evidence for key interactions between MSCs and other immune cells [17,18,19]. Although several “eat-me” signals were previously reported to drive phagocyte-mediated efferocytosis, the complexity of MSC surfactome combined with the bi-directional interaction with the surrounding environment insults suggest that it may be possible to “pre-select” sub-populations of MSCs with innate predisposition to efficiently inhibit unwanted inflammation. Amongst the long list of potential markers, we selected CD146, a receptor originally identified as a melanoma cell adhesion molecule [20]. Although CD146 is highly expressed in many tumors, endothelial cells and MSCs, recent evidence revealed that CD146 is not merely an adhesion molecule, as it can bind several ligands, including growth factors and extracellular matrixes [21]. We, thus, show, in this study, how pre-selection of an MSC sub-population based on high-CD146 expression defines a cellular biopharmaceutical capable of eliciting pronounced efferocytosis, consequently resulting in amplified immune suppression in the context of GVHD.

## 2. Materials and Methods

### 2.1. Animals and Ethics

All female C57BL/6 (6–8 weeks old or 33–36 weeks old for retired breeders) and Balb/c mice (6–8 weeks old) were purchased from the Jackson Laboratory (Bar Harbor, ME, USA) and housed in a pathogen-free environment at the animal facility located at the Institute for Research in Immunology and Cancer (IRIC). All experimental procedures and protocols were approved by the Animal Ethics Committee (CDEA) of Université de Montréal.

### 2.2. Antibodies and Reagents

The flow-cytometry antibodies (CD44, CD45, CD73, CD90, CD105, CD146, H2-K^b^, I-A^b^, CD107a and CXCR4) were purchased from BD Biosciences (San Jose, CA, USA). The CD146 neutralizing antibodies and the CellTrace^®^ reagent were purchased from Thermofisher Scientific (Markham, ON, Canada). The quantikines for murine interferon (IFN)-gamma and murine IL-10 were purchased from R&D Systems (Minneapolis, MN, USA). The indoleamine 2,3-dioxygenase (IDO)-1 ELISA was purchased from Cusabio Technology LLC (Houston, TX, USA). Recombinant murine IL-10 and IFN-gamma were purchased from Peprotech (Rocky Hill, NJ, USA). The Amicon Ultra-4 centrifugal filters were purchased from Millipore-Sigma (Burlington, MA, USA). Kits used for the isolation of B and T cells by positive selection were purchased from StemCell Technologies (Vancouver, BC, Canada).

### 2.3. Generation of BM-Derived MSCs

In order to generate BM-derived mouse MSCs, the femurs of 6–8-week-old female C57BL/6 mice were isolated and flushed with Alpha Modification of Eagle’s Medium (AMEM) supplemented with 10% FBS and 50 U/mL Penicillin–Streptomycin in a 10 cm cell culture dish, then incubated at 37 °C. Two days later, non-adherent cells were removed and the media replaced every 3 to 4 days until plastic-adherent cells reached 80% confluency. The generated cells were detached using 0.05% trypsin and expanded until a uniform MSC population was obtained. The generated MSCs were validated for their innate phenotype by flow-cytometry for the expression of CD44, CD45, CD73, CD90 and CD105. The cells were frozen in liquid nitrogen until use.

### 2.4. CD146 MSC Sorting, Phenotypic Analysis and Proliferation Analysis

To isolate CD146l^lo^ and CD146^hi^ MSCs, the parental (control—thereafter referred to as Ctl) population was first stained with anti-CD146 antibodies then the low or high 5% CD146-positive cells were sorted (indicated by the red arrows in Figure 1A) using the BD FACSAria Cell Sorter. Following their in vitro expansion, the phenotype of the MSC populations was validated using CD146 antibody prior to staining for H2-K^b^, I-A^b^, CD107a and CXCR4.

### 2.5. Cytokine and Chemokine Analysis

For cytokine and chemokine profiling, 15 cm cell culture dishes containing 80–90% confluent MSCs were grown in serum-free AMEM for 24 h at 37 °C and 5% CO_2_. Collected supernatants were then concentrated using the Amicon Ultra-4 centrifugal filters (3000 NMWL) for 1 h at 4 °C. Collected concentrates (80×) were then frozen at −80 °C until shipped to EveTechnologies (Calgary, AB, Canada) for cytokine/chemokine assessment by luminex. For IDO-1 quantification, the same approach was used in the context of untreated MSCs. To induce IDO-1 expression, MSCs were first stimulated with 5 ng/mL IFN-gamma for 24 h; the media was replaced with fresh serum-free media to be collected after 24 h as described above.

### 2.6. Two-Way Mixed Lymphocyte Reaction (MLR)

The two-way MLR is prepared by mixing 1 × 10^5^ C57BL/6-derived splenocytes with 1 × 10^5^ Balb/c-derived splenocytes (1:1 ratio) in a 96-well round-bottom plate. The cells were then incubated at 37 °C and 5% CO_2_ for 72 h prior to assessing IFN-gamma secretion in the supernatant by quantikine kit. For the co-culture assays, the two-way MLR was conducted on a layer of MSCs plated in a 48-well plate. For experiments involving the use of fixed MSCs, the cells were first plated the day before then fixed with 0.1% paraformaldehyde for 20 min at room temperature. After extensive but gentle washing (2–3X using sterile PBS), splenocytes were mixed then added on the layer of fixed MSCs. A similar approach was used to test the in vivo uptake of MSCs following in vitro CD146 neutralization by polyclonal antibodies.

### 2.7. Collection of Peritoneal Macrophages (pMACs)

Retired breeder C57BL/6 mice were first sacrificed prior to exposing their peritoneal cavity. A total volume of 20 mL serum-free RPMI was then injected in their peritoneum using a sterile syringe. After three lavages, the media were collected (~10–15 mL on average) and the cells were collected by centrifugation cycle of 10 min at 800× *g* and washed with PBS; this step was repeated twice. The obtained cells were then plated accordingly to the corresponding experiment protocol.

### 2.8. Allogeneic BM Transplantation

BALB/c recipient female mice were irradiated (8.5Gy) prior to transplantation by intravenous (IV) injection with 5 × 10^6^ C57BL/6-derived BM cells supplemented with 1 × 10^5^ purified CD3^+^ T cells isolated from the spleen of a C57BL/6 mouse. Mice were then assessed using an established scoring system. Briefly, the scoring was on a scale of 0–2 for each of the following parameters: posture, mobility, fur, skin rashes and weight loss. Once mice reached an average score of 2–3, 1 × 10^5^ of Ctl, CD146^lo^ or CD146^hi^ MSCs were injected intraperitoneally (IP). Mice where then monitored until reaching a score of 7–8 or showing a weight loss exceeding 20% [22].

### 2.9. Evaluating In Vivo Efferocytosis

To specifically assess in vivo efferocytosis, 10^6^ CellTrace^TM^-labelled C57BL/6-derived MSCs were IP-injected in immunocompetent 6–8-week-old female C57BL/6 mice (*n* = 3/group). Two hours later, injected mice were sacrificed and peritoneal lavage was conducted, as described above, using 20 mL of serum-free RPMI. Collected cells were centrifuged at 800× *g* for 10 min and the cell pellets washed twice with PBS. Recovered cells were then stained for CD11b^+^ and analyzed for their CellTrace^TM^ uptake by flow cytometry.

### 2.10. Statistical Analysis

*p*-values were calculated using one-way analysis of variance (ANOVA). Results are represented as average mean with standard deviation (S.D.) error bars and statistical significance is represented with asterisks: * *p* ˂ 0.05, ** *p* ˂ 0.01, *** *p* ˂ 0.001.

## 3. Results

### 3.1. Isolation and Phenotypic Characterization of CD146^hi^ MSCs

Prior to characterizing MSCs based on their CD146 expression levels, we first assessed the expression profile of CD146 on the ctl (parental) population by flow cytometry (Figure 1A—top panel). We next sorted two populations expressing low (^lo^) versus high (^hi^) CD146-expressing MSCs, respectively (red arrows—Figure 1A, Appendix A). Since CD146^hi^ MSCs were previously reported to exhibit higher cell surface levels of CD107a and CXCR4, we next assessed their expression on sorted MSCs by flow cytometry and found no major alterations (Figure 1B). When further assessed for MSC innate phenotypic markers, a lower CD44 expression level was detected on CD146^hi^ MSC, whereas CD105 levels were higher on the surface of the CD146^lo^ MSC subset (Figure 1C,D). Meanwhile, no changes were observed with respect to CD73 and CD90, and all MSC populations were negative for the hematopoietic marker CD45 (Figure 1C). In addition, all three MSC populations were positive for H2-K^b^ (MHCI) and remained I-A^b^ (MHCII) negative (Figure 1E), while PD-L1 was only detected on both CD146 sorted populations (Figure 1F). Overall, these results indicate that MSC separation based on CD146 expression levels defines sub-populations exhibiting variable cell surface markers.

**Figure 1 cells-11-02263-f001:**
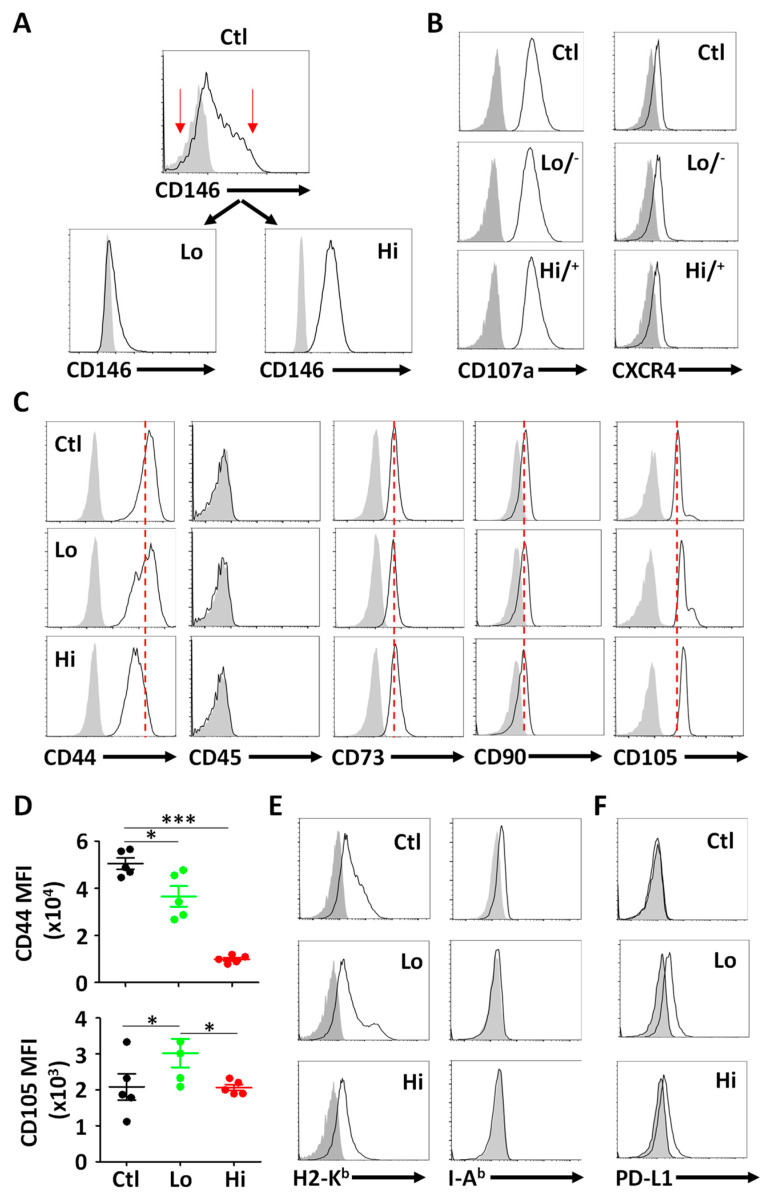
Isolation and phenotypic characterization of CD146^+^ MSCs. (**A**) Flow-cytometry analysis of CD146^lo^ versus CD146^hi^ MSCs following their sorting from the ctl parental MSC population (upper panel). (**B**) Flow-cytometry analysis of CD107a and CXCR4 expression on all three MSC populations. (**C**) Phenotypic analysis of the three different MSC populations according to ISCT guidelines. Isotype controls are shown as filled grey histograms. The red dotted line is placed according to the ctl parental MSC population. (**D**) Comparative analysis of CD44 and CD105 MFI on the different MSC populations. The ctl MSC population is displayed in black, CD146^lo^ MSCs in green and CD146^hi^ MSCs in red. (**E**) Flow-cytometry analysis of H2-K^b^ and I-A^b^ profiles of the three MSC populations. Isotype controls are shown in filled grey histograms. (**F**) Flow-cytometry analysis of PD-L1 profiles of the three MSC populations. Isotype controls are shown in filled grey histograms. For panel D, *n* = 5/group with * *p* < 0.05 and *** *p* < 0.01.

### 3.2. CD146^hi^ MSCs Exhibit Pronounced T-Cell Inhibition In Vitro

Based on the previously observed phenotypic differences, we next investigated whether separating MSCs based on CD146 expression affects their secretome profile. Interestingly, CD146^hi^ MSCs expressed high levels of interleukin (IL)-6, keratinocyte-derived chemokine (KC), monocyte chemotactic protein (MCP)-1, also known as CCL2, and vascular endothelial growth factor (VEGF) (Figure 2A). A significant increase was also observed in the production of lipopolysaccharide-induced CXC chemokine (LIX), granulocyte-colony-stimulating factor (G-CSF), granulocyte-macrophage colony-stimulating factor (GM-CSF) and macrophage inflammatory protein (MIP)-1alpha and MIP-2, which are all known potent chemoattractants for different subtypes of immune cells (Figure 2A). Macrophage colony-stimulating factor (M-CSF), on the other hand, was the only cytokine to be highly secreted by the ctl MSC population (Figure 2A). Although the addition of conditioned media derived from each of the MSC populations inhibited IFN-gamma production from a two-way MLR, the treatment group containing CD146^hi^ MSCs led to a more pronounced inhibitory effect (Figure 2B). This enhanced inhibition could not be attributed to IDO-1 secretion for the following reasons. First, IFN-gamma-stimulated ctl MSCs (parental population before sorting) secret the highest amount of IDO-1, yet their MLR inhibitory effect was moderate (Figure 2C). Second, both CD146^lo^ and CD146^hi^ MSCs secreted equivalent IDO-1 levels in response to IFN-gamma stimulation, which cannot explain the inhibitory differences observed in the MLR assay (Figure 2C). Finally, co-culturing the three MSC populations with the two-way MLR using different MSC-to-T-cell ratios (e.g., 1:1, 1:5, 1:10 and 1:20) revealed potent inhibitory effects with the CD146^hi^ group, as assessed by IFN-gamma quantification (Figure 2D). These results clearly indicate that CD146^hi^ MSCs exhibit a pronounced inhibitory effect on in vitro activated T cells.

### 3.3. CD146^hi^ MSCs Require Both Cell–Cell Contact and Soluble Factors to Inhibit Activated T Cells In Vitro

Despite an inhibitory effect observed with MSC-derived secretome, a more pronounced inhibition of IFN-gamma was observed when MSCs were co-cultured directly with activated T cells (Figure 2D). However, these studies could not depict whether CD146^hi^ MSCs rely solely on cell–cell contact and/or need soluble mediators to mediate T-cell inhibition. To investigate this question, two additional assays were conducted. In the first assay, MSCs were first killed by pre-fixation using paraformaldehyde to preserve cell surface molecules while impairing the cells’ ability to secrete soluble factors (Figure 3A). In contrast to the CD146^lo^ MSC treatment group, a two-fold decrease in IFN-gamma production was observed when activated T cells were co-cultured with pre-fixed CD146^hi^ MSCs (Figure 3B). The fact that IFN-gamma levels were not restored to a level comparable or close to a regular MLR response when the CD146^hi^ MSCs were pre-fixed suggests that cell–cell contact is required to mediate T-cell inhibition. To test that hypothesis, a second assay was designed to assess the role of CD146 in T-cell suppression. For this purpose, CD146^hi^ MSCs were first treated with a CD146 polyclonal antibody preparation prior to conducting the two-way MLR (Figure 3C). As anticipated, a two-fold increase in IFN-gamma was observed when CD146 was neutralized on the surface of live CD146^hi^ MSCs (Figure 3D). Altogether, these data clearly indicate that both soluble mediators and cell surface CD146 (amongst potentially other cell surface factors) are required for efficient inhibition of in vitro activated T cells by CD146^hi^ MSCs.

**Figure 2 cells-11-02263-f002:**
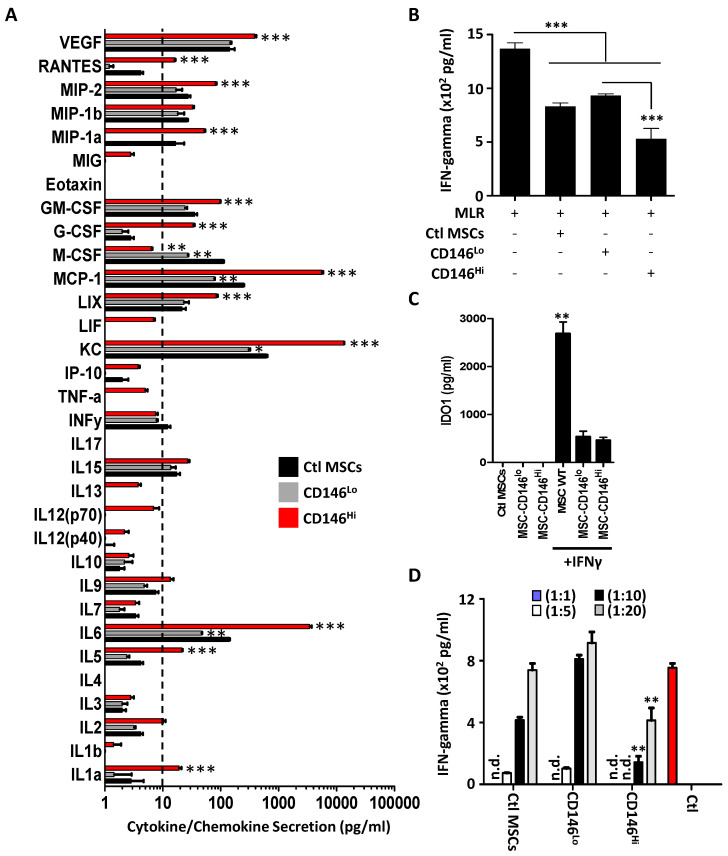
Assessment of the inhibitory properties of MSCs on activated T cells. (**A**) Luminex analysis of various cytokines and chemokines in the conditioned media derived from the ctl MSC population (black), CD146^lo^ MSC (grey) or CD146^hi^ MSC (red). For this experiment, *n* = 6/group with ** *p* < 0.01 and *** *p* < 0.001 compared to the ctl MSC population. (**B**) A two-way MLR using a mix of C57BL/6 and Balb/c splenocytes treated with the conditioned media collected from the different MSC populations. (**C**) IDO-1 quantification by ELISA using conditioned media derived from the three MSC populations with or without IFN-gamma pre-treatment. (**D**) A two-way MLR cultured on a layer of MSCs derived from ctl MSCs (parental), CD146^lo^ MSCs or CD146^hi^ MSCs at different MSC:T cell ratios. Ctl MLR (shown in red) consists of mixed splenocytes without MSCs. For all panels shown in this figure, *n* = 5/group with * *p* < 0.05, ** *p* < 0.01 and *** *p* < 0.001.

### 3.4. The Secretome of CD146^hi^ MSCs Does Not Trigger IL-10 Production from pMACs

MSCs were initially believed to exert their immune-suppressive capacity primarily via the secretion of soluble factors [23,24,25]. An elegant study by Giri et al. recently reported a CCL2–CXCL12 complex in the secretome of MSCs capable of inducing IL-10 production when in contact with endogenous macrophages [13]. We, thus, wondered whether treatment of pMACs using MSC-derived conditioned media could trigger IL-10 production (Figure 4A) and found that it was not the case (Figure 4B). We next tested whether other suppressive factors produced by conditioned-media-treated pMACs could potentially inhibit the two-way MLR (Figure 4C). As shown in Figure 4D, none of the tested conditions inhibited IFN-gamma production, suggesting that the initially observed in vitro T-cell inhibition with CD146^hi^ MSCs is mainly dependent on both MSC-derived soluble mediators and through a direct CD146 interaction.

### 3.5. CD146^hi^ MSCs Improve the Outcome of Mice with Acute GVHD through Enhanced In Vivo Efferocytosis

So far, our data allude to the fact that MSCs can suppress allogeneic T-cell activation. The inhibitory effect requires both soluble factors and direct contact of MSCs with target cells. To test their potency in an acute inflammatory model, the three MSC subsets were administered to Balb/c mice exhibiting acute GVHD following allogeneic BM transplantation (Figure 5A). Compared to untreated animals (black line), both ctl (blue) and CD146^lo^ MSC (green) groups suppressed inflammation and delayed death by 9 days (Figure 5B). Administration of CD146^hi^ MSCs (red line), on the other hand, led to a more pronounced therapeutic effect, with an overall 40% survival rate obtained up to 7 weeks post-BM transplantation (Figure 5B,C). Although one cannot preclude a direct T-cell inhibition mediated by injected MSCs, an alternative or complementary explanation to this therapeutic effect could involve efferocytosis mediated by CD11b^hi^ phagocytic cells, as previously shown by our lab in a study on thymoproteasome-expressing MSCs [26]. To test this hypothesis, CellTrace-labelled MSCs were next IP injected into naïve mice prior to conducting a peritoneal lavage 2 h post injection (Figure 5D). By gating on the three main CD11b populations collected from peritoneal lavage (Figure 5E), we found that all three MSC populations were, indeed, efficiently captured by CD11b^hi^ cells with a pronounced signal observed in mice treated with CD146^hi^ MSCs (Figure 5F—black arrow). Interestingly, in contrast to the remaining MSC populations, CD146^hi^ MSCs were also captured by CD11b^med^ cells (Figure 5F—red arrow). These data prompt us to investigate whether enhanced efferocytosis mediated by CD11b^hi^ myeloid cells could be impaired following CD146 neutralization. Indeed, pre-mixing CD146^hi^ MSCs with the anti-CD146 antibody prior to injection (Figure 5G) dramatically reduced their efferocytosis by CD11b^hi^ phagocytes compared to isotype-treated CD146^hi^ MSCs (Figure 5H). In summary, these data not only confirm our in vitro data demonstrating that CD146^hi^ MSCs are superior to their parental or CD146^lo^ MSC populations at inhibiting activated T cells, but they bring forward an important role for CD146 in directly stimulating efferocytosis by CD11b^hi^ myeloid cells.

## 4. Discussion

MSCs do not possess an intrinsic and specific set of defined markers. Thus, the objective of this study is not to shed light on the ambiguity of MSC phenotypes, but to potentially provide compelling evidence for additional, less-known signatures that may affect their immunomodulatory properties and perhaps provide a more robust response to pro-inflammatory insults than regular MSCs. We, thus, fractionated MSCs into two main sub-populations based on their CD146 expression profile. We selected the CD146 endothelial marker (also known as the melanoma cell adhesion marker) due to its established role in enhancing both cell–cell contact and migration [21,27]. We, indeed, provide compelling evidence that culture-adapted CD146^hi^ BM-derived MSCs represent a sub-population capable of robust suppression of activated T cells, both in vitro and in the context of GVHD. In addition, our data strongly allude to a novel function for CD146 as a “rheostat”, regulating the cross-talk between MSCs and myeloid cells. As such, the CD146 marker could be exploited as a “Trojan horse” capable of delivering a given MSC-derived effect through enhanced in vivo efferocytosis (graphical abstract).

The use of MSC-based cell therapy for the treatment of catastrophic illnesses has grown exponentially following the survival of a patient with grade IV GVHD treated with haploidentical MSCs [28]. Since then, hundreds of clinical trials were initiated to treat all types of autoimmune diseases [29]. Although MSCs are known to possess potent suppressive effects, their exact mode of action remains a matter of debate. For instance, several studies have demonstrated that MSCs can convert pro-inflammatory M1 macrophages to an anti-inflammatory M2 phenotype, reprogrammed to produce less tumor necrosis factor-alpha and nitric oxide, while increasing their IL-10 secretion levels [30]. However, recent reports proposed a new mechanism, whereby injected MSCs must undergo apoptosis prior to exerting their suppressive effects [16,31,32]. If so, how can infused MSCs continue exerting a long-term regulatory function despite their in vivo clearance? If we assume that this mechanism is behind the survival extension observed in the GVHD model for all MSC-treated groups, then the pronounced effect of CD146^hi^ MSCs could potentially be due to two independent factors. First, expression of high CD146 levels on the surface of MSCs could lead to enhanced contact with target T cells, which may explain their potent inhibitory effect when tested in the context of an MLR. This, however, does not preclude the potential effect from soluble mediators, as shown in our in vitro assay. An example would be the increased production of IL-6 by CD146^hi^ MSCs, which has been largely associated with the suppressive phenotype of MSCs in general, as it plays important roles in inhibiting pro-inflammatory peripheral mononuclear cells and blocking the dendritic cell maturation process [33]. Second, CD146 is an adhesion molecule and, thus, can enhance cell–cell contact, resulting in superior efferocytosis by myeloid/phagocytic cells. Although our in vivo experiment highlights an important new role for CD146 as a novel “eat-me” signal, follow-up studies could shed more light on how low versus high CD146-expressing MSCs reprogram endogenous phagocytes as a means to transmit the MSC suppressive message to activated T cells. In addition, the therapeutic potential of the secretome derived from CD146^hi^ MSCs can be further explored to study the role of extracellular vesicles or exosomes as a means to treat ailments in a cell-free approach [34]. 

## 5. Conclusions

Further elucidations into cellular and molecular interactions mediated by CD146^hi^ MSCs and the myeloid compartment will certainly better inform future investigations on key cellular and molecular events that may ultimately bridge the gaps to advance the clinical use of MSCs. Fractionating MSCs based on CD146 expression may not affect the innate ability of MSCs to respond to surrounding pathophysiological cues, but would rather allow the use of a cellular product endowed with a natural capacity to suppress or orchestrate cellular and molecular changes in target phagocytic cells required to restore immune balance.

## Figures and Tables

**Figure 3 cells-11-02263-f003:**
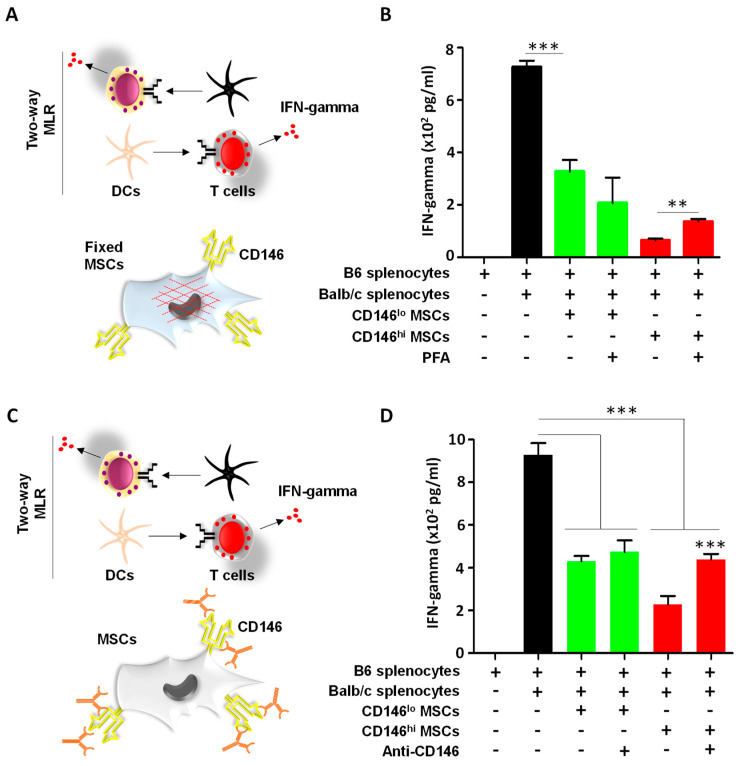
CD146^hi^ MSCs require both soluble and cell–cell contact to inhibit activated T cells. (**A**) Schematic diagram showing the design of the inhibitory experiment performed using pre-fixed MSCs. (**B**) IFN-gamma quantification of the experiment depicted in panel A. (**C**) Schematic diagram showing the design of the inhibitory experiment performed using the anti-CD146 neutralizing polyclonal antibodies. (**D**) IFN-gamma quantification of the experiment depicted in panel C. For panels B and D, *n* = 5/group with ** *p* < 0.01 and *** *p* < 0.001.

**Figure 4 cells-11-02263-f004:**
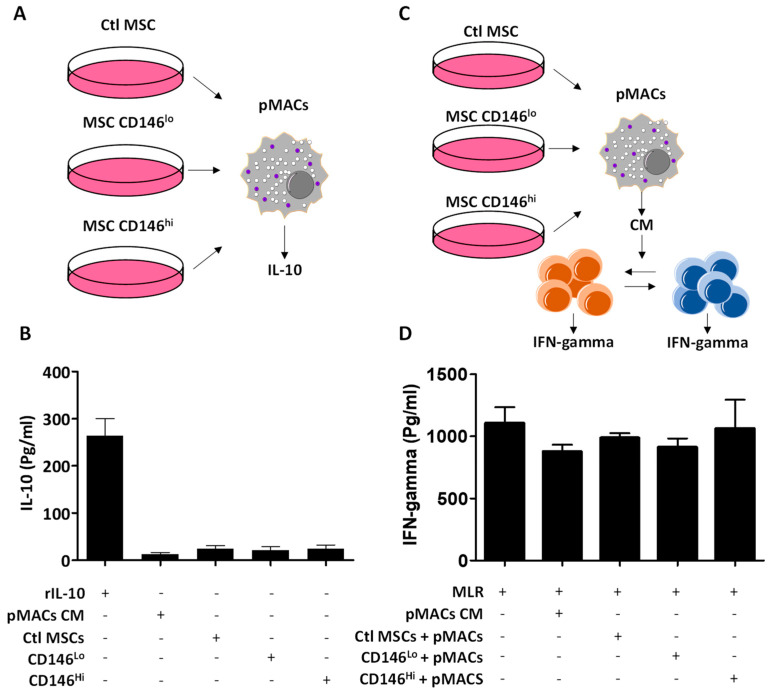
The inhibitory effect of MSCs is independent of pMAC-derived soluble mediators. (**A**) Schematic diagram showing the experimental design used to assess IL-10 production from pMACs treated with MSC-derived conditioned media. (**B**) IL-10 quantification for the experiment shown in panel A. (**C**) Schematic diagram showing the experimental design used to assess IFN-gamma production from a two-way MLR treated with pMACs cultured with MSC-derived conditioned media. (**D**) IFN-gamma quantification from the experiment shown in panel D. For panels B and D, *n* = 5/group.

**Figure 5 cells-11-02263-f005:**
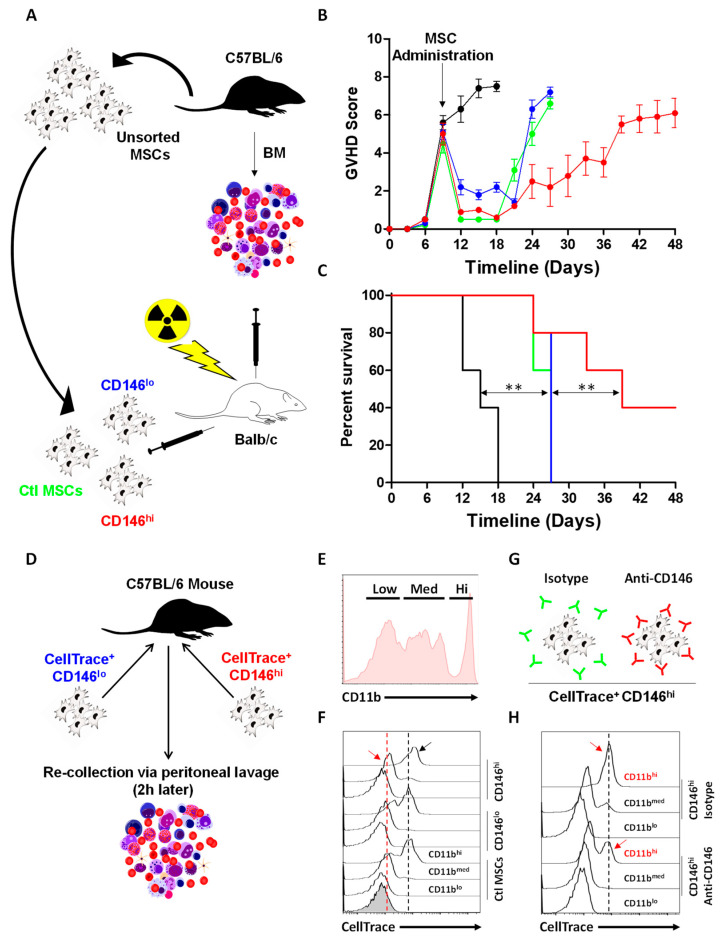
CD146^hi^ MSC administration to GVHD mice improves their therapeutic outcome. (**A**) Schematic diagram of the acute GVHD study design. (**B**) GVHD mouse scoring following administration of the different MSC populations. The ctl acute GVHD (no MSCs) is depicted in black, parental MSCs in green, CD146^lo^ MSCs in blue and CD146^hi^ MSCs in red. (**C**) Kaplan–Meier survival curve of mice suffering from acute GVHD and treated with the different MSC populations. (**D**) Schematic diagram of the design used to assess in vivo efferocytosis of MSCs. (**E**) Representative flow-cytometry analysis of CD11b expression profile on cells collected following peritoneal lavage. (**F**) Analysis of CellTrace on CD11b^+^ cells collected 2 h following MSC administration. (**G**) Schematic diagram of the antibody treatment of CD146^hi^ MSCs prior to their in vivo injection. (**H**) Assessment of CellTrace on CD11b^+^ cells derived from mice treated with cells as shown in panel G. For this panel, *n* = 5–10/group with ** *p* < 0.01.

## Data Availability

Not applicable.

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
