# Peer review of "CD146 Defines a Mesenchymal Stromal Cell Subpopulation with Enhanced Suppressive Properties"

_cells, 2022, doi:10.3390/cells11152263_

Round 1

Reviewer 1 Report

Broad comments

The paper is extremely interesting, the research design is very appreciable and the results are original and widely useful for next research and translational medicine studies.

My opinion about the manuscript is very good, but some comments are given for trying to improve at higher level this manuscript.

Major comments

1.     Figure 2 A. Can you discuss about IL-2 higher production in CD146hi MSC? This is involved in T-cell proliferation.

2.     Figure 2A. the legend says *P<0.05 and ***P<0.01. compared to what?

3.     Figure 2A, the legend lacks of indication of ** (p-value?)

4.     Figure 4A and 4C. By comparing them it should suggested to remove CM in 5C or add it also in 5A. Not?

5.     Figure 3B can be marked with green and red edging lines for the bars indicating CD146low and CD146hi, thus to help the reader to compare the filled green and red bars in panel Figure 3D

6.     In Figure 5A the cartoon of the mouse is obviously repeated, but I think it will be more clear to do not replicate the mouse since it is the same irradiated Balb/c, instead it can be confused with a different graft receiving mouse. One mouse with radioactive icon can be enough.

7.     Figure 5F can be improved, maybe making similar to 5H for labeling histogram lines with CD11-level

8.    May the authors briefly discuss how their results on the CD146 will be important for the next generation of cell-free stem cell derived biofactors, such as expression and role in exosomes and microvesicles? Please look at the article describing the critical role of selection of stem cells for this aim doi.org/10.3390/antibiotics10070750

9.     In the discussion section the authors discuss signatures that may affect their immunomodulatory properties, I would like to highlight, suggest the reading and to report a paper located  at doi: 10.3390/jcm11051236 that cites the role of membrane fatty acid signature in improving immunomodulatory function and angiogenic differentiation capacity of fetal membrane MSCs, as well as the critical role in other stem cells properties.

and other adult stem cells

Minor comments

10.  “In sum“ at Line 290, did they mean “in summary”?

11.  Please check sentence at line 279 . “To test this hypothesis, CellTrace-labelled MSCs we next IP injected to naïve mice prior to conducting a peritoneal lavage 2h post-injection (Fig. 5D)”

Author Response

REVIEWER 1

Comment #1

Figure 2 A. Can you discuss about IL-2 higher production in CD146hi MSC? This is involved in T-cell proliferation.

Response to comment #1

We thank the reviewer for his/her comment. Although we perfectly agree with the reviewer on the importance of IL-2 in supporting T-cell proliferation, we chose not to discuss that point specifically as the amount secreted is negligible (9.34 + 1.13 pg/ml for CD146hi versus 4.2 + 0.717 pg/ml for the parental MSC population). Thus, specific interpretation of this data may induce the scientific community in error by highlighting a pivotal increase/role for this cytokine. In addition, we did not observe any unusual or enhanced T-cell proliferation during the co-culture studies indicating again no major role played by IL-2 in this context. 

Comment #2

Figure 2A. the legend says *P<0.05 and ***P<0.01. compared to what?

Response to comment #2

It is compared to the ctl MSC population. This information was clarified in the figure legend (highlighted in yellow).

Comment #3

Figure 2A, the legend lacks of indication of ** (p-value?)

Response to comment #3

We thank the reviewer for pointing-out this mistake, which was corrected. In fact, the values should be **P<0.01 and ***P<0.001.

Comment #4

Figure 4A and 4C. By comparing them it should suggested to remove CM in 5C or add it also in 5A. Not?

Response to comment #4

In fact, no! Experiment depicted in Figure 4A was to assess IL-10 production by macrophages in response to MSCs conditioned-media (CM). Since no IL-10 was detected there (Fig. 4B), we next assessed if other suppressive factors were secreted by MSC-derived CM-treated pMACs, and found no effect there to.

Comment #5

Figure 3B can be marked with green and red edging lines for the bars indicating CD146low and CD146hi, thus to help the reader to compare the filled green and red bars in panel Figure 3D.

Response to comment #5

The figure modification was modified as requested by reviewer 1.

Comment #6

In Figure 5A the cartoon of the mouse is obviously repeated, but I think it will be clearer to do not replicate the mouse since it is the same irradiated Balb/c, instead it can be confused with a different graft receiving mouse. One mouse with radioactive icon can be enough.

Response to comment #6

Figure 5A was modified as requested by reviewer 1.

Comment #7

Figure 5F can be improved, maybe making similar to 5H for labeling histogram lines with CD11-level

Response to comment #7

We thank the reviewer for his/her suggestion. However, we feel that keeping the figure that way is better as the read-outs are different and the figure might become confusing to understand.

Comment #8

May the authors briefly discuss how their results on the CD146 will be important for the next generation of cell-free stem cell derived biofactors, such as expression and role in exosomes and microvesicles? Please look at the article describing the critical role of selection of stem cells for this aim doi.org/10.3390/antibiotics10070750

Response to comment #8

We totally agree with the reviewer that additional studies could be conducted to decipher the therapeutic potential of the secreatome derived from these MSC sub-populations. Thus, the following section was added at the end of the discussion to highlight this point:

"In addition, the therapeutic potential of the secreatome derived from CD146hi MSCs can be further explored to study the role of extracellular vesicles or exosomes as a means to treat ailments in a cell-free approach."  

Comment #9

In the discussion section the authors discuss signatures that may affect their immunomodulatory properties, I would like to highlight, suggest the reading and to report a paper located at doi: 10.3390/jcm11051236 that cites the role of membrane fatty acid signature in improving immunomodulatory function and angiogenic differentiation capacity of fetal membrane MSCs, as well as the critical role in other stem cells properties, and other adult stem cells

Response to comment #9

We thank the reviewer for his/her comment. However, we feel that discussing the impact of membrane fatty acid on the immune-modulatory properties of these MSCs is out of scope for this study. This topic can however be part of a different study to be conducted in the near future.

Comment #10

"In sum" at Line 290, did they mean “in summary”?

Response to comment #10

The term "In sum" was replaced by "In summary".

Comment #11

Please check sentence at line 279. “To test this hypothesis, CellTrace-labelled MSCs we next IP injected to naïve mice prior to conducting a peritoneal lavage 2h post-injection (Fig. 5D)”

Response to comment #11

We thank the reviewer for his/her comment. The sentence had a type at "we next IP", which should have been written as "were next IP". The correction was made.

Reviewer 2 Report

The manuscript is written clearly and has used high-quality methods, however, in its current status, the manuscript has low novelty and doesn't sound original. The discussion and big picture of the work are poorly presented.

There are several published papers describing the role of CD146 expression on MSCs and immunomodulation of T cells, macrophages, and release of IL-6, TGFbeta, IDO, and IL-10. The authors need to demonstrate clearly and specifically, how their work is original and novel compared to these previous works. This can be included throughout the paper and particularly abstract/introduction/discussion. Is the novelty around the GVHD model or efferocytosis or...?

The results and abstract indicate that CD146 hi cells express pro-inflammatory cytokines such as IL-6 and TNF while also these cells show increasing IL-10 and suppress T cells. How authors can explain these conflicting findings in the context of inflammation, GVHD, or autoimmune diseases?

Some methods are not clear and need citation referred to the chosen method. Why 24 hours were used to collect samples for secretory proteins +/- IFN-y. Compared to similar work,  3-5 days will be needed to collect these data. Also, why MSC fixation with formaldehyde was done?

In figure 1, the flow-cytometry image showing CD105 expression (Fig 1C) is not presenting the findings of CD105 is less in CD146hi than CD146low (Figure 1D).

Discussion should indicate how GVHD mechanisms can be reversed by CD146hi MSCs.

Author Response

REVIEWER 2

Comment #1

There are several published papers describing the role of CD146 expression on MSCs and immunomodulation of T cells, macrophages, and release of IL-6, TGFbeta, IDO, and IL-10. The authors need to demonstrate clearly and specifically, how their work is original and novel compared to these previous works. This can be included throughout the paper and particularly abstract/introduction/discussion. Is the novelty around the GVHD model or efferocytosis or...?

Response to comment #1

We thank the reviewer for his/her comment. We perfectly agree that CD146 has been previously associated with enhanced immunomodulation of specific immune cells. However, our study brings forward two important novelties. First, it clearly shows that CD146hi MSCs have higher suppressive abilities in the context of mixed lymphocyte reactions (in vitro) and GVHD (in vivo). In other words, these cells can be specifically used in this indication following allogeneic or mismatched bone marrow/stem cells. Second, we are providing evidence that CD146 could serve as a new "eat me" signal that may enhance the suppressive ability of these MSCs in the context of an inflammatory reaction. In fact, we are currently using MSCs as a delivery vehicle for cancer antigens while exploiting the idea that CD146hi MSCs could serve as better vectors to deliver these antigens to myeloid and antigen presenting cells.

The end of the abstract and parts of the results section already stated such information as follow:

"Altogether, this study provides the impetus to adopt an isolation protocol for MSCs based on CD146 expression profile before their therapeutic use, and suggests, a major role played by CD146 as a novel "eat-me" signal capable of enhancing MSC uptake by competent phagocytes."      

"In summary, these data not only confirm our in vitro data demonstrating that CD146hi MSCs are superior to their parental or CD146lo MSC populations at inhibiting activated T cells, but it brings forward an important role for CD146 in directly stimulating efferocytosis by CD11bhi myeloid cells."

Comment #2

The results and abstract indicate that CD146 hi cells express pro-inflammatory cytokines such as IL-6 and TNF while also these cells show increasing IL-10 and suppress T cells. How authors can explain these conflicting findings in the context of inflammation, GVHD, or autoimmune diseases?

Response to comment #2

We thank the reviewer for his/her comment. In fact, this might seem confusing at a first sight, but one has to pay close attention to the used scale in Figure 2A. As such, detected TNF is considered negligible (4.6 + 0.5 pg/ml), which is why we put a dotted line to discriminate between what we consider below a threshold level. So, everything below that line is considered background or of low relevance. IL-6, on the other hand, has been associated with the suppressive phenotype for MSCs. For instance, it plays important roles in inhibiting pro-inflammatory PMNs and can block the DC maturation process (F van den Akker et al. Mediators Inflamm. 2013). Therefore, the increase observed in this cytokine is central to the inhibitory properties of these MSCs.      

Comment #3

Some methods are not clear and need citation referred to the chosen method. Why 24 hours were used to collect samples for secretory proteins +/- IFN-y. Compared to similar work, 3-5 days will be needed to collect these data. Also, why MSC fixation with formaldehyde was done?

Response to comment #3

In fact, we have previously compared the effect of IFN-y on MSCs after 12, 24 and 48h and noticed no major differences between the 24 and 48h. We have therefore selected this timeline and used it consistently in all of our studies. As for MSC fixation with formaldehyde, we wrote the following sentence in the MS:

"MSCs were first killed by pre-fixation using paraformaldehyde to preserve cell surface molecules while impairing the cell’s ability to secrete soluble factors (Fig. 3A)."

In other words, we wanted the MSCs to be metabolically dead while preserving any cell surface protein (example CD146) to discriminate between the effects mediated by cell-cell contact versus the secreatome.

Comment #4

In figure 1, the flow-cytometry image showing CD105 expression (Fig 1C) is not presenting the findings of CD105 is less in CD146hi than CD146low (Figure 1D).

Response to comment #4

We thank the reviewer for pointing out this mistake. The text was modified to highlight the higher expression of CD105 on CD146lo MSCs.

Comment #5

Discussion should indicate how GVHD mechanisms can be reversed by CD146hi MSCs.

Response to comment #5

The following section in the discussion already highlights the possible mechanisms by which CD146hi MSCs inhibit GVHD:

The use of MSC-based cell therapy for the treatment of catastrophic illnesses has grown exponentially following the survival of a patient with grade IV GVHD treated with haploidentical MSCs [28] . Since then, hundreds of clinical trials were initiated to treat all types of autoimmune diseases [29]. Although MSCs are known to possess potent suppressive effects, their exact mode of action remains a matter of debate. Recent reports proposed a new mechanism whereby injected MSCs must undergo apoptosis prior to exert their suppressive effects [16, 30, 31]. If we assume that this mechanism is behind the survival extension observed in the GVHD model for all MSC-treated groups, then the pronounced effect of CD146hi MSCs could potentially be due to two independent factors. First, expression of high CD146 levels on the surface of MSCs could lead to enhanced contact with target T cells, which may explain their potent inhibitory effect when tested in the context of a MLR. This however does not preclude the potential effect by soluble mediators as shown in our in vitro assay. Second, CD146 is an adhesion molecule, thus, can enhance cell-cell contact resulting in superior efferocytosis by myeloid/phagocytic cells. Although our in vivo experiment highlights an important new role for CD146 as a novel "eat-me" signal, follow-up studies could shed more light on how low versus high CD146-expressing MSCs re-program endogenous phagocytes as a mean to transmit the MSC suppressive message to activated T cells.

Reviewer 3 Report

this manuscript deals with a relevant mechanism for the "in vivo" action of mesenchymal stromal cells. 

Description of efferocytosis and the subsequent effects should be better described. Moreover, "in vitro" may not reproduce the "in vivo"  conditions. Authors should be careful not to overclaim results, especially when local factors may be involved

Author Response

REVIEWER 3

Comment #1

Description of efferocytosis and the subsequent effects should be better described.

Response to comment #1

The discussion has been modified to clarify this point as follow:

"The use of MSC-based cell therapy for the treatment of catastrophic illnesses has grown exponentially following the survival of a patient with grade IV GVHD treated with haploidentical MSCs [28] . Since then, hundreds of clinical trials were initiated to treat all types of autoimmune diseases [29]. Although MSCs are known to possess potent suppressive effects, their exact mode of action remains a matter of debate. For instance, several studies have demonstrated that MSCs can convert pro-inflammatory M1 macrophages to an anti-inflammatory M2 phenotype reprogrammed to produce less Tumor necrosis factor-alpha and nitric oxide while increasing their IL-10 secretion levels [30]. However, recent reports proposed a new mechanism whereby injected MSCs must undergo apoptosis prior to exert their suppressive effects [16, 31, 32]. If so, how can infused MSCs continue exerting a long-term regulatory function despite their in vivo clearance?"

Comment #2

Moreover, "in vitro" may not reproduce the "in vivo" conditions. Authors should be careful not to overclaim results, especially when local factors may be involved.

Response to comment #2

We perfectly agree with the reviewer on that point. This is why we introduced nuances throughout the text by saying that MSCs depend on cell-cell contact, their secreatome and efferocytosis, with the latter point being central to their mode of action.

Reviewer 4 Report

In this manuscript, the authors showed that two separate MSC subpopulations, CD146hi and CD146low, possess distinct secretome profile. Through further in vitro studies, the authors demonstrated that CD146hi subpopulation of MSCs could inhibit T cell activation and the important role of CD146 in eliciting efferocytosis by CD11b myeloid cells. Finally, the authors showed that CD146hi MSC population could improve therapeutic outcome in GVHD mice. While the concept of CD146 as playing an immune regulatory function is interesting, several major issues needed to be addressed before the manuscript can be published.

MAJOR ISSUES

1)      The foundation of this study is solely based on sorting of CD146hi and CD146low from parental MSCs. Separation of CD146hi and low population is based on high or low 5%. The rationale of this strategy remains unclear and needs further in text clarification. In Figure 1, the authors should also include dot plots, as histograms do not provide information on cell subpopulations/clustering. Do the CD146hi and low populations show distinct separation?

2)      Gating strategy for flow cytometry and cell sorting should also be included as supplementary methods.

3)      In Figure 2C, WT MSCs (presumably the same as Ctl MSCs) expressed more than 2500pg/ml IDO1 after IFNγ treatment. CD146low and CD146hi appeared to show 500pg/ml or less in each group. That combined to be only around 1000pg/ml or less. Can the authors clarify the disparity? Were all the groups seeded the same number of cells?

4)      The BM microenvironment is where MSCs most likely interacting with BM MACs, rather than pMACs. There are also distinct morphological and phenotypic differences between BM MACs and pMACs. Authors should demonstrate impact of these CD146hi/low MSCs on BM MACs.

https://bmcimmunol.biomedcentral.com/articles/10.1186/1471-2172-14-6

MINOR ISSUES

1)      Line 65: “not merely and” should be “not merely an”.

2)      Line 133: typo for “protocol”.

3)      Figure 3B and Figure 3D: The order of groupings should be consistent.

Author Response

REVIEWER 4

Comment #1

The foundation of this study is solely based on sorting of CD146hi and CD146low from parental MSCs. Separation of CD146hi and low population is based on high or low 5%. The rationale of this strategy remains unclear and needs further in text clarification. In Figure 1, the authors should also include dot plots, as histograms do not provide information on cell subpopulations/clustering. Do the CD146hi and low populations show distinct separation?

Response to comment #1

We thank the reviewer for his/her comment. In fact, the selection of top or low 5% was arbitrary to be able to work with sub-populations expressing different levels of CD146. We could have selected the top/low 1 or 10%. There is no specific rationale for that point except of having two distinct populations. Having said that, it would be indeed interesting to test in the future whether top/low 1, 5 and 10% exhibit similar effects in terms of immune-suppression.

As for the dot blot request, we included these data as a separate supplementary figure 1.  

Comment #2

Gating strategy for flow cytometry and cell sorting should also be included as supplementary methods.

Response to comment #2

A supplementary figure 2 has been prepared to show the gating strategy used for the flow cytometry and sorting of these cells.

Comment #3

In Figure 2C, WT MSCs (presumably the same as Ctl MSCs) expressed more than 2500pg/ml IDO1 after IFNγ treatment. CD146low and CD146hi appeared to show 500pg/ml or less in each group. That combined to be only around 1000pg/ml or less. Can the authors clarify the disparity? Were all the groups seeded the same number of cells?

Response to comment #4

This is indeed an important point. First, we would like to confirm that indeed, the same cell number was seeded for all conditions. As for the observed disparity, we have to remember that we have isolated two small sub-fractions (representing 10%) from the parental MSC population. The only possible explanation in this case is that most of the high IDO-1 expressers are found in between these two sub-fractions. We cannot rule-out at the moment idiosyncratic effects that may be caused by the sorting process. A definitive answer could be however provided using a single-cell RNA-seq approach, which is beyond the scope of this study.

Comment #4

The BM microenvironment is where MSCs most likely interacting with BM MACs, rather than pMACs. There are also distinct morphological and phenotypic differences between BM MACs and pMACs. Authors should demonstrate impact of these CD146hi/low MSCs on BM MACs.

https://bmcimmunol.biomedcentral.com/articles/10.1186/1471-2172-14-6

Response to comment #4

We totally agree with the reviewer on the likely interaction between MSCs and bone marrow-resident MACs. However, there is a distinction to make in this case as we are infusing culture-adapted MSCs back into the animal via the intraperitoneal route. This means that MSCs are directly interacting with MACs of the peritoneal cavity. This is also clearly highlighted with the in vivo efferocytosis study displayed in panel 5. This is why we focused on testing the effect of MSCs on pMACs instead of bone marrow-derived MACs.

Comment #5

Line 65: “not merely and” should be “not merely an”.

Response to comment #5

We wish to thank the reviewer for pointing-out this typo. The correction has been made.

Comment #6

Line 133: typo for “protocol”.

Response to comment #6

We wish to thank the reviewer for pointing-out this typo. The correction has been made.

Comment #7

Figure 3B and Figure 3D: The order of groupings should be consistent.

Response to comment #7

The order is indeed maintained.

Round 2

Reviewer 2 Report

The authors covered most of comments, but,

response to comment 2 should be added to the discussion.

Author Response

As requested by reviewer 2, a sentence was added to the discussion to highlight the effect of IL-6.

Reviewer 4 Report

Based on the reviewers' responses and changes, I am happy to accept in its current form. I propose one more change to be made to improve the manuscript,

Figure 3D- First group (black bar) should be swapped with the second group so it is consistent with Figure 3B.

Author Response

We thank the reviewer for pointing-out this error. The black bar has been swapped as requested.